# Hydrogels: Properties and Applications in Biomedicine

**DOI:** 10.3390/molecules27092902

**Published:** 2022-05-02

**Authors:** Tzu-Chuan Ho, Chin-Chuan Chang, Hung-Pin Chan, Tze-Wen Chung, Chih-Wen Shu, Kuo-Pin Chuang, Tsai-Hui Duh, Ming-Hui Yang, Yu-Chang Tyan

**Affiliations:** 1Department of Medical Imaging and Radiological Sciences, Kaohsiung Medical University, Kaohsiung 807, Taiwan; r090340@kmu.edu.tw (T.-C.H.); t0987295916@gmail.com (C.-W.S.); 2Department of Nuclear Medicine, Kaohsiung Medical University Hospital, Kaohsiung 807, Taiwan; chinuan@kmu.edu.tw; 3School of Medicine, Kaohsiung Medical University, Kaohsiung 807, Taiwan; 4Neuroscience Research Center, Kaohsiung Medical University, Kaohsiung 807, Taiwan; 5Department of Electrical Engineering, I-Shou University, Kaohsiung 840, Taiwan; 6Department of Nuclear Medicine, Kaohsiung Veterans General Hospital, Kaohsiung 813, Taiwan; hpchan@vghks.gov.tw; 7Biomedical Engineering Research and Development Center, National Yang Ming Chiao Tung University, Taipei 112, Taiwan; twchung@nycu.edu.tw; 8Graduate Institute of Animal Vaccine Technology, College of Veterinary Medicine, National Pingtung University of Science and Technology, Pingtung 912, Taiwan; kpchuang@g4e.npust.edu.tw; 9Department of Medicinal and Applied Chemistry, Kaohsiung Medical University, Kaohsiung 807, Taiwan; tshudu@kmu.edu.tw; 10Research Center for Environmental Medicine, Kaohsiung Medical University, Kaohsiung 807, Taiwan; 11Department of Medical Education and Research, Kaohsiung Veterans General Hospital, Kaohsiung 813, Taiwan; 12Center of General Education, Shu-Zen Junior College of Medicine and Management, Kaohsiung 821, Taiwan; 13Graduate Institute of Medicine, College of Medicine, Kaohsiung Medical University, Kaohsiung 807, Taiwan; 14Department of Medical Research, Kaohsiung Medical University Hospital, Kaohsiung 807, Taiwan; 15Center for Cancer Research, Kaohsiung Medical University, Kaohsiung 807, Taiwan

**Keywords:** hydrogel, medical application, 3D cell culture, drug delivery, wound dressing, tissue engineering

## Abstract

Hydrogels are crosslinked polymer chains with three-dimensional (3D) network structures, which can absorb relatively large amounts of fluid. Because of the high water content, soft structure, and porosity of hydrogels, they closely resemble living tissues. Research in recent years shows that hydrogels have been applied in various fields, such as agriculture, biomaterials, the food industry, drug delivery, tissue engineering, and regenerative medicine. Along with the underlying technology improvements of hydrogel development, hydrogels can be expected to be applied in more fields. Although not all hydrogels have good biodegradability and biocompatibility, such as synthetic hydrogels (polyvinyl alcohol, polyacrylamide, polyethylene glycol hydrogels, etc.), their biodegradability and biocompatibility can be adjusted by modification of their functional group or incorporation of natural polymers. Hence, scientists are still interested in the biomedical applications of hydrogels due to their creative adjustability for different uses. In this review, we first introduce the basic information of hydrogels, such as structure, classification, and synthesis. Then, we further describe the recent applications of hydrogels in 3D cell cultures, drug delivery, wound dressing, and tissue engineering.

## 1. Introduction

Hydrogels comprise a three-dimensional (3D) network which can absorb a large amount of water and swell in the water due to their hydrophilic groups, such as -NH_2_, -COOH, -OH, -CONH_2_, -CONH, and -SO_3_H [1,2,3,4,5,6,7,8,9]. Its network is usually constructed by crosslinked polymer chains that sometimes can be formed through crosslinked colloidal clusters [10,11,12,13,14,15,16,17]. They can be flexible and soft, which are results of their water absorption ability [18]. Chemical or physical crosslinking of natural or synthetic polymer chains can be used to design the hydrogels [19,20,21,22,23]. Because of the high water content, soft structure, and porosity of hydrogels, they closely resemble living tissue. Wichterle and Lim first developed hydrogels for biomaterials in 1960. They produced a synthetic poly-2-hydroxyethyl methacrylate (PHEMA) hydrogel, which was then used as a filler for eye enucleation and contact lenses [24]. Since then, the expense of hydrogels in drug delivery and bioactive compound release has been elevated in several early studies from the 1970s to the 1990s [25,26,27,28,29]. In the 1990s, hydrogels were applied in tissue engineering [30,31,32,33]. The application of hydrogels was restricted to only the surface environment from the 1970s to the 1990s, for applications in the eye or open wounds, for example. The properties (e.g., swelling–deswelling rate, stiffness, degradability, mech size) of hydrogels can be adjusted by changing the hydrophilic and hydrophobic ratios, the initiator or polymer concentrations, and the reaction conditions (time, temperature, container, etc.) [34,35,36,37]. The biomedical application of hydrogels is not limited to the surface environment due to in situ gelation after infection and the stimuli responsiveness of the hydrogel [38,39].

Over the past 60 years, hydrogels have been engineered to be implantable, injectable, and sprayable for many organs and tissues [38,39]. Recently, hydrogels have gained attention in the field of environmental engineering [40], soft robotics [41], and wastewater treatment [42]. With the underlying technological improvement of hydrogel generation, hydrogels can be expected to be used in more fields. However, scientists are still interested in biomedical applications of hydrogels, as evidenced by 25,000 references to hydrogels for biomedical applications in the past five years (Figure 1). In this review, we first introduce the basic information on hydrogels, such as their structures, classification, and synthesis. Then, we further describe the recent biomedical applications of hydrogels. We mainly emphasize the biomedical applications of hydrogels in 3D cell cultures, drug delivery, wound dressings, and tissue engineering.

## 2. Classification and Structure of Hydrogels

The classification of hydrogels depends on their source, composition, environmental stimuli, crosslinking, property, configuration, and ionic charge [43,44,45,46,47,48,49,50,51,52,53,54,55,56,57,58,59,60,61,62,63,64], as briefly shown in Figure 2. Hydrogels are formed by the crosslinking of polymer chains. The sources of hydrogels can be divided into natural, synthetic, or semi-synthetic polymers. Within the polymer source, hydrogels can be termed natural, synthetic, or semi-polymer hydrogels. Naturally derived hydrogels (natural hydrogels) include cellulose, chitosan, collagen, alginate, agarose, hyaluronic acid, gelatin, and fibrin, etc. [43,44]. They have inherent biocompatibility, bioactivity, and biodegradability, but relatively weak stability and mechanical strength. Although natural hydrogels are safe for the majority of the population, some materials of natural hydrogels are allergens in rare cases [65,66,67,68,69,70]. Thus, natural hydrogels have potential immunological risks if they are used in the treatment of sensitive individuals. Synthetic hydrogels are constructed by synthetic polymers. Those polymers are human-made polymers, which are prepared through the polymerization of a monomer, such as polyvinyl alcohol (PVA), polyethylene glycol (PEG), polyethylene oxide (PEO), poly-2-hydroxyethyl methacrylate (PHEMA), poly-N-isopropyl acrylamide (PNIPAM), polyacrylic acid (PAA), and polyacrylamide (PAAM) [43,44,45]. Although a few of them are biocompatible, such as PAAM, they are stable and have mechanical strength. Semi-synthetic polymers are chemically modified natural polymers or a combination of natural and synthetic polymers as materials for the preparation of semi-hydrogels. An example of chemically modified natural polymers is methacryloyl-modified gelatin (GelMA) [71], or acrylate-modified hyaluronic acid (AcHyA) [72]. Additionally, it can be a combination of natural and synthetic polymers, such as PEG-conjugated fibrinogen, or gelatin and albumin [73]. These hydrogels not only present the bioactivity features of natural hydrogels but also have multi-tunable properties through their diverse chemical parameters [74].

The polymers in the hydrogels can be homopolymers, copolymers, semi-interpenetrating networks (semi-IPNs), or IPN hydrogels, which are dependent on their composition. The simple diagrams of homopolymer hydrogels, copolymer hydrogels, semi-IPNs, and IPN hydrogels are shown in Figure 3. The polymer chains of homopolymer hydrogels are derived from one species of monomer, whereas copolymer hydrogels are derived from two or more species of monomer [46,47]. The copolymer can be further classified as block, alternate, or random copolymer, based on their composition order of monomer (Figure 3). The active side of copolymers can be linked with another monomer or copolymer. Both homo- and copolymer hydrogels contain one type of polymer chain. In contrast, both semi-IPNs and IPN hydrogels have two or more types of polymer chains. Semi-IPN hydrogels are a polymer network embedded in the linear polymer chains. The linear polymer chains are embedded without a crosslinking agent. IPN hydrogels are formed by two or more polymer networks that are crosslinked with each other by using a crosslinking agent [48,75]. In comparison with the homo- and copolymer hydrogels, the semi-IPN and IPN hydrogels present higher mechanical strength and swelling properties [76,77,78,79].

Hydrogels can be amorphous, crystalline, or semi-crystalline based on their configuration. The amorphous hydrogels are random network structures at the molecular level. Crystalline hydrogels consist of a tightly packed polymer network structure with the order of crystallization. Semi-crystalline hydrogels were developed by chemical crosslinking in 1994, which contains the amorphous and crystalline regions [49,50,51,52]. After this time, physical crosslinked semi-crystalline hydrogels have been prepared through bulk and micellar polymerization [80,81,82,83]. Semi-crystalline physical hydrogels rapidly change from a solid-like state to a liquid-like state and are reversible compared with semi-crystalline chemical hydrogels, which are extensively applied in preparing injectable hydrogels and shape memory hydrogels [82,83].

Based on the crosslinking method, the hydrogels can be classified into chemical hydrogels and physical hydrogels. Chemical hydrogels have permanent junctions composed of covalent crosslinking and polymerizing end-functionalized macromeres. Physical hydrogels have a transient junction comprising physical interactions, such as ionic interactions, hydrogen bonding, and crystallization [53]. Thus, the mechanical properties of physical hydrogels are poorer than chemical hydrogels due to weak physical interaction. Relatively, physical hydrogels are soft and have reversibility from liquid to solid [84].

Chemical hydrogels have permanent junctions composed of covalent crosslinking and polymerizing end-functionalized macromeres of polymer chains. Thus, hydrogels can be divided into four groups, namely nonionic, anionic, cationic, and ampholytic, based on their ionic charge. Apart from nonionic hydrogels, the remaining hydrogels contain electric charges in their polymer chain and are pH-sensitive due to their ionic groups. The anionic hydrogels usually have negative electric charges in their polymer chain, whereas cationic hydrogels have positive electric charges. The ampholytic hydrogels contain both negative and positive electric charges through copolymerization of anionic and cationic monomers or incorporation of zwitterionic monomers into the polymer network [54,55,56]. These ionic hydrogels can form a complex with other molecules having electronic charges and become applicable in drug delivery for treatment of disease [85,86].

The response of hydrogels can be physical, chemical, or biomedical. It has been revealed that physical, chemical, or biomedical stimulations adjust the physical properties of hydrogels, such as deformation (a transformation between the swollen hydrogel and shrunken hydrogel) and self-assembly [57,58,59,60,61,62,63,64]. Those stimulations exist in external solvents or environments. The physical stimulations include temperature, pressure, light, electric field, and magnetic fields [58,59]. Chemical stimulations contain pH, ionic strength, solvent composition, and molecular species [58,60,61,62]. Biomedical stimulations involve the response of antigen, ligand, and enzyme [61,62,63,64]. Along with the responsive extent of physical properties, the hydrogels are divided into conventional hydrogels and smart hydrogels based on their properties. Conventional hydrogels have only a small alteration consisting of swelling with external environmental conditions and low mechanical strength, whereas smart hydrogels are sensitive to the small changes of external environmental conditions, and immediately adjust their physical properties (such as mechanical strength, swellability, and stimuli-sensitivity) [57,58,59,60,61,62,63,64].

The solid portion of a hydrogel is a 3D network structure of crosslinked polymer chains [87,88,89,90]. It is usually referred to as a mesh at the molecular level, containing mesh size and crosslinking of polymer chains (Figure 4). The crosslinking of hydrogels can be formed by the covalent (chemical) or physical (junction/entanglement) linking. Supramolecular polymers are a new category of polymers that can potentially be used for biomaterial applications beyond the limitations of conventional polymers. The molecular weight of supramolecular hydrogels may not be infinite, because their crosslinking is a series of specific but nonpermanent interactions [91,92,93,94]. They are polymeric arrays of monomeric units held together by reversible and highly directed secondary interactions, which are noncovalent bonds, such as the hydrogen bonding interaction, π-π stacking, and the host–guest interaction. Thus, the resulting materials retain their polymeric properties in solution.

As seen in Figure 4, the covalent bond can be linked between two functional groups of polymers with or without covalent agents (under enzyme catalysis). The physical junction can be formed by the electronic interactions between the opposite charges in the polymer. It also can be linked with hydrogen bonds or ionic interactions with the ions in solution [87,88]. The hydrogel mesh holds the water and has the elastic force that can be caused by the swelling and release of water [89]. Therefore, the meshes can maintain the solidity of hydrogels [89]. They are important for the exchange of fluids across the polymer network and carrying the cell or drug [90]. The size of the mesh holes is referred to as mesh size, which is associated with the release time of drug delivery [95] and correlated with the linear distance between two crosslinking points (ξ) [95,96]. Demeter M et al. used the rheological analysis to demonstrate that the average molecular weight between crosslinking points (Mc) and crosslink density (Ve) affects the ξ of mesh size. There is a positive correlation between Mc and mesh size [97]. Steinman NY et al. demonstrated that 4 kDa and 8 kDa PEG can form 11 kDa and 16 kDa hydrogels at the same concentration of a crosslinking agent [98]. Thus, the molecular weight of hydrogels is basically dependent on the content of polymers, which can be infinite due to their 3D network structure [98,99].

## 3. Synthesis of Hydrogels

Hydrogels are crosslinked polymer chains with a three-dimensional (3D) network structure. The crosslinking in the hydrogels can happen not only after the polymer chain is synthesized, but also with the growth of the polymer chain. Thus, the synthetization of hydrogels begins with monomers, prepolymers, and polymers [87]. Regardless of material type, crosslinking in hydrogels can be physical or chemical. Methods of crosslinking hydrogels are shown in Figure 5. The physical crosslinking in hydrogels includes hydrogen bonding, amphiphilic graft, block polymers formation, crystallization, ionic interactions, and protein interactions. [100,101,102,103,104,105,106,107,108,109,110,111,112]. The hydrogen bonding is present between molecules containing N-H, O-H, or F-H functional groups. Thus, the polymer with the functional groups as above can form the hydrogels through hydrogen bonding. The amphiphilic graft and block polymer formation means that polymers can self-assemble in hydrophobic or hydrophilic solvent due to their amphiphilic affinity. With crystallization, the polymer chains can synthesize the hydrogels through adjustment of their crystallized temperature. The freeze–thaw and heating process is one of the common methods of crystallization. The ionic interactions occur in crosslinking via ionic group attraction. Protein interaction occurs in the polymers, which are added or modified with protein by molecular medical technology, such as protein and genetic engineering. Those polymers can synthesize the hydrogels through antibody–antigen interactions or the properties of the protein (crystallization, functional groups, or hydrogen bonding) [113,114,115,116].

Methods of chemical crosslinking include chemical reaction, high-energy radiation, free-radical polymerization, and enzymes (Figure 5). The chemical reaction occurs in complementary or pendant groups or polymers with a crosslinking agent. Both high-energy radiation and free-radical polymerization for hydrogel preparation occur through free-radical crosslinking. Free-radical production for high-energy radiation occurs via gamma ray or an electronic beam; free-radical polymerization occurs via enzyme catalysts or UV excitation.

Enzymes catalyzed by crosslinking happen in those polymers which are modified or incorporated with enzyme-sensitive molecules. Recently, some new methods have been used to prepare hydrogels. For example, the photon–Fenton method uses sunlight for hydrogel production, which can be achieved without any crosslinking agents for the preparation of biodegradable polyethyleneimine nanogels. Another self-assembly method can form hydrogels through specific and local interactions of polymers.

## 4. Biomedical Applications of Hydrogels

### 4.1. 3D Cell Cultures

Three-dimensional cell cultures provide a useful platform for the cell to grow in vitro in all directions. Compared with the 2D culture system, it is easier to understand the in vivo cell behavior, since cells form a 3D structure in living tissue. The 3D cell culture is achieved by culturing the cells on a 3D scaffold. In the in vivo 3D cell structures, the cells are embedded in the extracellular matrix (ECM) and form a 3D structure. ECM is known to play an important role in regulating the cell behavior [117]. Hydrogels have a 3D structure and a hydrophilic polymer network capable of absorbing water in addition to biological fluid [1,2,3,4,5,6,7,8,9,118,119]. Thus, they can construct the soft and wet 3D structure which is like the extracellular matrix (ECM), which is available to encapsulate the cells. This results in those hydrogels which have gained increasing attention in the application of scaffolds for 3D cell cultures [120,121].

Hydrogels can comprise natural, synthetic, and semi-synthetic polymers. These hydrogels provide distinct biochemical, physical, and mechanical properties for the 3D cell culture [117]. Table 1 describes the recent application of these hydrogels for 3D cell culture. Natural hydrogels have good biocompatibility, endogenous factors, and the similar viscoelasticity and fibrils of the ECM. These hydrogels can support cell activity for 3D cell cultures.

Collagen hydrogels are natural hydrogels. Collagen is the most abundant fibrous protein in the human body. It is also one of the major structural elements of the ECM, which responds by providing elastic strength and regulating cell adhesion, chemotaxis, migration, and tissue development [122,123]. In addition, it plays a major role in the function of articular and bone tissue [120]. Due to the biofunction of collagen in human body, the collagen hydrogels can mimic the properties and characteristics of organisms [124]. Previous studies have shown that collagen can improve cell growth, adhesion, differentiation of neural cells [124], and form tissue-like structures of chondrocyte for co-culturing cells, such as neural cells and chondrocytes [125]. Both in vivo and in vitro systems indicate that collagen can be significant in maintaining the chondrocyte phenotype and sustaining chondrogenesis [126]. It attracts the attention of collagen hydrogels in the 3D cell culture of the chondrocyte. Jin G.Z. and Kim H.W. have demonstrated that type I collagen maintains the chondrogenic phenotype, by 3D culturing the rat chondrocyte in type I collagen hydrogel scaffolds [127]. Other studies further demonstrate that, when the type II collagen hydrogel scaffold is compared to type I collagen, the type II collagen can significantly enhance the chondrogenic differentiation of human mesenchymal stem cells (hMSCs) [128]. Although the type I and type II collagen support the chondrogenic phenotype via different mechanisms, the collagen hydrogel scaffold presents a potential ability for cartilage tissue engineering [128,129]. Kilmer C. E et al. found that the rat mesenchymal stem cells (rMSCs)-encapsulated collagen type I/II hydrogels scaffold facilitates the chondrogenic differentiation of rMSCs and presents a statistically higher cartilage repair ability in the cartilage defective rat model. This result indicates the clinical value of collagen hydrogels in cartilage repair [130].

The denseness of capillary networks can facilitate the nutrition of cells, which is important for generating the natural tissue in vitro. Many studies have attempted to use a combination of endothelial cells (ECs) and natural or synthetic materials to reconstruct the microvasculature of natural tissue in vitro [131,132]. The shortcoming of this construction is that the interference of fetal bovine serum (FBS) cannot be avoided for future in vivo applications [133]. Andree B et al. demonstrate that the collage hydrogel scaffold, under serum-free conditions, can promote human umbilical vein endothelial cells (HUVECs) to develop more of the EC network when combined with human-adipose-tissue-derived stromal cells (hASCs) [134]. Nonetheless, long-term performance of type I collage may be affected by significant shrinkage and weak mechanical properties [135,136]. Buitrago O. J et al. improved the poor mechanical properties and cell-mediated shrinkage of collagen hydrogel by hybrid silk fibroin with collagen. This collagen–silk fibroin hydrogel scaffold can promote cell viability and cell growth of hMSCs in long-term 3D cultures [136].

Hyaluronic acid (HA) is a source of natural hydrogels. It is also a major structural element of ECM and is widely distributed in many tissues, such as skin and cartilage [137]. HA can facilitate cellular survival, migration, angiogenesis, differentiation, and neural regeneration via transducing the intracellular signals [138,139,140]. Wu S et al. found that HA hydrogel scaffolds can promote the neural differentiation of human-induced, pluripotent-stem-cells-derived neural progenitor cells (hiPSC-NPCs) [141]. Another study used a microencapsulation approach to encapsulate the hiPSCs in the HA-rich core–shell hydrogel microcapsules. This 3D culture system improves the cell quality and efficient cardiac differentiation of hiPSCs [142]. Ren Y et al. has synthesized the stiffness of adjustable HA hydrogels for 3D culturing of rMSCs. This HA hydrogel scaffold can maintain the stem characteristics of rMSCs and induce direct cartilage differentiation [143]. These studies show the potential effects of HA hydrogel in application of stem cell therapy and tissue repair and regeneration.

In addition, HA is overexpressed in different types of cancers. The higher content of HA in the cancer microenvironment supports tumor progression and anticancer drug resistance [144,145]. Suo A et al. prepared an HA hydrogel scaffold through a hydrazone and photo dual crosslinking process. This HA hydrogel scaffold had similar topography and mechanical properties to the ECM of breast cancer tumors in vivo. Breast cancer MCF-7 cells were 3D cultured in this scaffold, and the migration/invasion abilities and tumorgenicity of these cells are greater than in 2D culture cells [146]. Other researchers report that HA hydrogel scaffolds promote the adhesion and proliferation of the human hepatocellular carcinoma cell line HepG2. Their results also show that the HA hydrogel scaffold enhances the activity of multidrug resistance proteins of HepG2 cells and reduces the therapeutic effectiveness in comparison to the 2D cultured cells [147]. These studies reveal the value of HA hydrogels in cancer research.

Another interesting application of HA hydrogel is to improve the efficacy of 3D cell cultures by mixing these materials with various substances [117]. An example is the microbial-transglutaminase-crosslinked HA semi-IPN with chondroitin, which enhances the osteogenic potential of dental pulp stem cells [148]. A cell-adhesive, peptide-RGD-modified HA hydrogel scaffold with high cell adhesion ability can increase the differentiation of human neural stem/progenitor cells (hNS/PC) toward oligodendrocytes and neurons in 3D cell cultures. This also shows that RGD-peptide-modified HA hydrogels support the long-term cell viability of hNS/PC with minimal growth factors [149]. Lou J et al. has reported that HA–collagen hydrogels will mimic the viscoelasticity and fibrillary of ECM. This hydrogel promotes the cell spreading, fiber remodeling, and focal adhesion of hMSCs in 3D cell culture [150].

Fibrin is also a major protein in constructing ECM, in addition to collagen and HA, which are formed by fibrinogens [151]. Fibrin hydrogel presents bioactivities, such as angiogenesis and a cell–matrix interaction through fibrin clotting, which have been used for 3D cell culture of adipose-derived stem cells and cardiomyocytes [152,153]. Recent studies of fibrin hydrogels are associated with mimicking a native ECM matrix and evaluating its effect on 3D cell cultures. Heo D.N et al. mimicked a native ECM matrix by incorporating the collagen in fibrin hydrogels. The collagen/fibrin hydrogels can induce the prevascular formation of HUVECs, which further improves the cell viability and proliferation of hMSCs and promotes their osteogenic differentiation and bone mineral deposition [154]. Gorczyca G et al. employed an alginate/fibrin IPN hydrogel which can form a gas and nutrition exchange space as a native ECM. They found that this hydrogel can encapsulate the porcine cumulus–oocyte complexes (COCs) and maintain the functional relationship between oocytes and follicular cells. They provided a novel culture platform for in vitro maturation of oocytes, which is important in reproductive biotechnology [155]. Bachmann B et al. adjusted the stiffness of fibrin hydrogels toward the elasticity of native chondrocyte ECM by using different concentrations of fibrin. Their study showed that the fibrin hydrogel under 30 kPa Young’s best modulus can induce the production of glycosaminoglycans and collagen type II from 3D-cultured primary human chondrocytes; it is best to trigger chondrocyte redifferentiation [156]. Another study also used a similar method to mimic the elasticity of the bone marrow niche. Their results show that fibrin hydrogel prepared under lower pressure as a 3D scaffold is the best option for enhancing the murine hematopoietic stem/progenitor cells’ (mHPSCs) expansion and differentiation [157]. However, the fast biodegradability of fibrin hydrogels results in an unstable 3D cell culture environment. Jarrell D. K. et al. found that hydrogels in high salt conditions can slow the degradation rate compared with that of fibrin hydrogels in physiologic-salt conditions. This hydrogel did not affect the viability and prevascular formation of encapsulated cells, such as human dermal fibroblasts, hiPSCs, and HUVECs [158].

Alginate is present in the cell wall of brown algae. Monomers of brown-algae-derived alginate can crosslink to form natural hydrogels [159]. Recent studies focus to improve the interaction of alginate for mammalian cells by modifying the alginate with cellular molecules [160,161]. Hunt C. N et al. utilized RGD modified alginate hydrogel to a 3D culture of the human embryonic stem cells to induce pluripotent stem cells (hESCs/hiPSCs). Their result indicates that RGD-modified alginate hydrogel can enhance the generation of retinal pigmented epithelium and neural retina from 3D-cultured hESCs/hiPSCs [162]. Moxon R. S et al. report that alginate–collagen hydrogels can enhance the cell adhesion of hiPSCs-derived neurons and promote the formation of complex neuron networks in a 3D culture model [163]. These results suggest that cellular-molecule-modified alginate hydrogels have a good cell adhesion property for the application of 3D cell cultures.

Synthetic hydrogels have good mechanical strength to provide structural support for various cell types in 3D cell cultures. These PVA and PEG hydrogels are widely used for the 3D scaffold due to its biocompatibility (adaptability and nontoxicity for living tissue), absence of immunogenicity, and adjustable stiffness [164,165]. Wilkinson et al. report that PVA hydrogels can replace the albumin to enhance the expansion of murine hematopoietic stem cells (mHSCs) [166]. PVA hydrogels with growth factors can promote the differentiation rate of mouse spermatogonia stem cells (mSCCs) into meiotic and post-meiotic cells [167]. In addition, several human glioma cell lines (LN299, U87MG, and Gli36) can form the tumor spheroids in 3D cultures with PVA-hydrogels-coated cell plates [168]. A similar result was also found in the 3D culture of human breast cancer Hs578T cells and human pancreatic cancer cell lines (Sui67 and Sui72) with PVA hydrogels. This result further indicates that PVA hydrogels can reduce cancer cell apoptosis and promote cancer cellular proliferation [169].

PEG hydrogels have been applied to the formation of multicellular tumor spheroids by 3D cell cultures for anticancer drug screening [170]. PEG hydrogels are also used for 3D cultures with stem cells to evaluate their efficacy in hiPSCs differentiation and the behavior of mMSCs at similar stiffness of myocardial infarction microenvironments in vivo [171,172]. In addition, PEG, by itself or combined with polycaprolactone, has good potential to encapsulate diverse bioactive molecules for lowering complicating factors derived from hydrogels [173,174]. Another increasing application of PEG hydrogels is to encapsulate the bioactive factors to promote cellular functions for 3D cultured cells. Research of de Sousa Araújo E et al. describes synthesis of the in situ chondroitin-sulfate-crosslinked PEG hydrogels through the Diels–Alder click reaction. This hydrogel can prolong the oxygen release of chondrocytes under 3D cultures. It may maintain the oxygen level in the articular cartilage and support the differentiation, viability, and proliferation of chondrocytes [175]. Jansen L. E et al. developed PEG hydrogels, which are separately incorporated with different combinations of bone-marrow-specific and cell-instructive peptides. This hydrogel also mimics the stiffness of the ECM in the bone marrow. The results show that this hydrogel can be used to evaluate the effect of different commotions in cell behaviors of hMSCs in the bone marrow microenvironment in vitro [176]. Overall, synthesis of hydrogels may have a potential value in the medical application of stem cells, restoration of infertility, anticancer drug screening, and repair of articular cartilage.

Although both natural and synthetic hydrogels have widely displayed their advantages in the 3D cell cultures, there are some disadvantages to those hydrogels. Fast degradation, weak stability, and poor mechanical strength are the disadvantages of natural hydrogels; while synthetic hydrogels are biologically inert and lack endogenous factors. Semi-synthetic hydrogels comprise a combination of natural and synthetic polymers [71]. The combination improves the individual disadvantages of natural and synthetic hydrogels in 3D cell cultures. For example, HA-PEG is a semi-synthetic hydrogel. It expresses a similar microenvironment to in vivo ECM, which promotes viability and functionality of hepatocytes derived from hiPS [177]. It also can promote the human umbilical vein endothelial cell (HUVECs) spheroids to form the capillary-like sprouts [178]. Nam S et al. have increased the stress–relaxation rate of RGD–alginate hydrogels by crosslinking with PEG. This hydrogel promotes fibroblasts to proliferate and distribute in addition to enhancing osteogenic differentiation of mMSCs [179]. Compared with natural and synthetic hydrogels, semi-synthetic hydrogels create a cellular environment that is closer to those in vivo. Further evaluation is needed to determine whether semi-synthetic hydrogels can mimic tissue performance.

### 4.2. Drug Delivery

Polymers are one of the most promising substances for the preparation of drug delivery systems. Polymers can be prepared for various nanostructures, including polymeric micelles, polymeric vesicles, and hydrogels. Those nanostructures are great for drug delivery [180]. Increased interest in hydrogels is focused on smart hydrogels due to the stimuli-responsive properties of polymeric moieties. Stimuli-responsive properties can enable the formulation of novel targeted drugs and control drug release through non-intravenous administration. It can also delay the effect of opsonization by low blood contact.

The basic advantage is the ability of a smart hydrogel to change its properties (such as mechanical properties, swelling capacity, hydrophilicity, or permeability of bioactive molecules) under the effect of surroundings, including temperature, pH, electromagnetic radiation, magnetic field, and biological factors. Smart hydrogels can be prepared by the natural or synthetic polymers. There is a problem with natural hydrogels, which is that its mechanical properties make it difficult to maintain consistency.

Although this problem with natural hydrogel can be overcome by extensive chemical modification for natural polymers, it is very difficult to process [181]. In contrast, synthetic polymers are easy to alter their chemical or physical properties. The biodegradable and hydrophilic synthetic polymers are the most competitive substances for the synthesis of smart hydrogels for drug delivery. Those synthetic polymers endow smart hydrogels with low toxicity, low side effects, and low blood material adhesion. Of these, the low blood material adhesion can slow the effect of opsonization and reduce the phagocyte elimination. Table 2 shows the application of those synthetic smart hydrogels for drug delivery.

Under temperature changes, the matrix volume of thermo-responsive hydrogels (TRHs) can be changed due to expansion or contraction, and the solubility, conformation, and phase transition of the polymer may change. The character of this hydrogel can be maintained in the form of gel at a range of temperatures. Zhao G et al. found that a prodrug, N-2-hydroxypropyl methacrylamide (HPMA)-copolymer-based dexamethasone, can change its shape from a liquid form at 4 °C to a hydrogel form at 30 °C or over through increasing the content level of Dex. This HPMA-Dex can become a hydrogel (ProGel-Dex) after intraarticular administration in rodent models of inflammatory arthritis and osteoarthritis. ProGel-Dex can stay in the joint for more than 30 days and release the water-soluble polymetric prodrug. Free-Dex is released through phagocytic synoviocytes. It can persistently improve joint inflammation and pain in rodents with inflammatory arthritis and osteoarthritis. Their results also show that the molecular weight of a water-soluble polymetric prodrug is very low (6.8 kDa), which ensures rapid renal clearance. It avoids the risk of the potential side effects of glucocorticoids [182]. Xing R et al. prepared topotecan (chemotherapeutic-drug-loaded) lipid nanoparticles (TPT-SLNs), and then incorporated TPT-SLNs into poloxamer 407 and poloxamer 188 solutions to form TPT-SLNs-TRHs. The gelation temperature of TPT-SLNs-TRHs is over 31 °C. They further evaluated the anticancer efficacy of TPT-SLNs-TRHs in the colorectal tumor xenograft rat model. Their results show that TPT-SLNs-TRHs can enhance antitumor efficacy and reduce the toxicity effect in comparison with the administration of pure TPT and TPT-SLNs. In addition, the antitumor effect of TPT-SLNs-TRHs can be stationary for 28 days. The stability tests show that particle size and incorporation efficiency of TPT-SLNs-TRHs are not changed over a period of six months. Whether the antitumor effect of TPT-SLNs-TRHs can exceed 28 days needs to be studied further [183].

Scientists are trying to apply the TRHs to deliver drugs for infectious diseases. Lamivudine (3TC) and zidovudine (AZT) are two compounds used for HIV treatment and prevention of AIDS progression. Patients require frequent dosing of these two compounds to achieve successful therapy for AIDS. Excessively frequent doses can reduce adherence, which may result in treatment failure. Witika B. A et al. used the Pluronic^®^ F-127 THSs to embed the nano co-crystal 3TC and AZT drug (NCC-3TC-AZT). The in vitro release data show that the release time of 3TC and AZT from the NCC-3TC-AZT-THSs is over 168 h. Cell toxicity results show that this hydrogel does not affect cell viability in comparison with the treatment of AZT and 3TC alone. Although the antiviral effect of NCC-3TC-AZT requires further examination, their results suggest that THRs have the potential of drug carriers for AIDS treatment [184].

Sustained release is being explored to increase the plasma and tissue residence time of polymer–protein therapeutics, thereby improving efficacy. Huynh V et al. have used hydrophilic polyethylene glycol methyl ether methacrylate (PEGMA) polymers to replace the PEG for preparation of the antibody-conjugated THRs. This hydrogel can control the release rate of proteins by adjustment of the content level of PEGMA. These THRs in the gel at room temperature can slowly dissolve over 37 °C to release the antibody. An in vitro assay shows that this hydrogel can completely release the protein within 13 days. The results suggest that PEGMA THRs can persistently release antibodies as carriers [185].

pH-responsive hydrogels can swell depending on the change of pH value in the environment. During the swelling period, the interior structure of pH-responsive hydrogels contains the absorbed water, resulting in the embedded drugs being released. This hydrogel can be used to release the drug in the stomach or intestine through oral administration [186]. Due to the fact that tumor environments are acidic, the pH-responsive hydrogels can be a drug carrier for antitumor therapy [187]. Qing W et al. replaced the OTs of methoxy PEG (mPEG)-OTs with luteolin (LUL) to obtain the mPEG-LUL. Bortezomib (BTZ), a new chemotherapy drug for colorectal cancer, is conjugated with LUL in mPEG-LUL by a borate ester bond to form mPEG-LUL-BZT. This hydrogel can sustain and release BTZ at pH 5.5 for up to 50 h. In vivo tumor elimination shows that intravenous administration of mPEG-LUL-BZT significantly reduces the tumor size in the colorectal tumor xenograft mouse model and is not toxic for normal mice. In addition, they also incorporate the indocyanine green (ICG) (a photothermal agent) into mPEG-LUL-BZT to prepare the pH–photo dual responsive hydrogel for colorectal cancer treatment with a combination of photothermal (PTT)/photodynamic (PDT) therapy. The result shows that the antitumor efficacy of mPEG-LUL-BZT-ICG is higher than mPEG-LUL-BZT. Their results suggest that mPEG-LUL is suitable to deliver BZT and enhance the specific killing of BZT for colorectal tumors. This hydrogel can also enhance its tumor therapeutic ability with a combination of photothermal agents [188].

Lin X et al. has developed a pH-responsive hydrogel, based on methacrylic acid copolymer (MAC) and polycaprolactone (PCL), by esterification reaction. MAC-g-PCL becomes gelatinous at pH 1.2 and dissolves at pH 7.4. They further embed the radioprotective agent, amifostine (S-2(3-aminopropylamino) ethyl phosphorothioate (Ami), to prepare the drug-carried hydrogel (MAC-g-PCL-Ami). An in vitro cell assay shows that MAC-g-PCL-Ami is out of the cytotoxicity range. The release profile of MAC-g-PCL-Ami shows that a small amount of Ami is released by stimulation with gastric fluid (pH 1.2), but a burst is released by stimulation with intestinal fluid (pH 7.4). Oral administration of MAC-g-PCL-Ami can protect the irradiated mice from hematopoietic acute radiation syndrome and extend their survival. Their results suggest the MAC-g-PCL pH-responsive hydrogel can prevent the degradation of Ami at the stomach and enhance effective delivery in the intestine. This hydrogel may be a potential carrier for drug with poor oral activity [189].

Photoresponsive hydrogels can alter their properties by using light energy. The alteration can be controlled easily by switching the light on and off, in a wavelength-specific manner, and regulating times of light stimulation [190,191]. Ghani M et al. developed a photoresponsive hydrogel by using the carboxylated spiropyran (SPCOOH)-modified silicone (poly(HEMA-co-PEGMEA)) IPNs. The light-induced drug release mechanism relies on the isoform switching of SPCOOH in the SPCOOH-modified silicone (poly(HEMA-co-PEGMEA)) IPNs. Their results show that doxycycline release from this hydrogel can be controlled by the UV light, which is compared to the hydrogels without SPCOOH modification. In addition, this hydrogel also reduces the early burst release of doxycycline. This hydrogel also can slowly release doxycycline up to 42 h during the UV stimulation [192]. However, there is a lack of cell or animal experiments to confirm the release profile and safety of this hydrogel in vivo.

Diabetic patients present low medication adherence, since they need to administer a self-subcutaneous injection of insulin. However, the microcarrier for subcutaneous drug delivery rarely achieves even distribution of the drug. It further restricts drug release and absorption. Fan L et al. has loaded the photoresponsive hydrogel microsphere into microcarrier-integrated microneedles to develop photoresponsive hydrogel microcarrier-integrated microneedles. The photoresponsive hydrogel microsphere contains the black phosphorus (BP) and poly (N-isopropylacrylamide) (pNIPAM), which are generated by using a flexible capillary microfluidic method. This photoresponsive hydrogel microsphere has the ability for photothermal energy conversion and maintains the bioactivity of encapsulated drugs. The microneedle is prepared by the mixture of porous ethoxylated trimethylolpropane triacrylate (ETPTA) and PEG. This microneedle can provide the mechanical strength to support the skin penetration and to deliver drugs uniformly under the skin. Their results show that photoresponsive hydrogel microcarrier-integrated microneedles can control the insulin release to manipulate the level of blood glucose in streptozotocin (STZ)-induced diabetic mice [193]. Their study suggests that photoresponsive hydrogel-microcarrier-integrated microneedles may increase the medication adherence and promote the uniform distribution of drug.

Apart from responsive hydrogel with a single stimulation, recent studies have developed dual-responsive hydrogels for drug delivery. Abedi F et al. prepared a noncytotoxic pH-/thermo-responsive hydrogel by free-radical crosslinking of N-Isopropylacrylamide (NIPAAm, thermo-responsive polymer) and *N*,*N*-dimethyl-aminoethyl methacrylate (DMAEMA, pH-responsive polymer). This poly (NIPAAm-co-DMAEMA) pH-/thermo-responsive hydrogel can persistently and simultaneously release the doxorubicin (DOX) and chemo-sensitizer curcumin (CUR) at pH 5.8/40 °C for 168 h. Their data also show that the DOX/CUR-poly (NIPAAm-co-DMAEMA) pH-/thermo-responsive hydrogel significantly induces the cell apoptosis of the HT-29 colon cancer cell line in comparison to the free DOX and CUR alone [194]. The poly (NIPAAm-co-DMAEMA) pH-/thermo-responsive hydrogel is also used to encapsulate the anticancer drug methotrexate (MTX). The MTX can be released up to 50 h at pH 5.5/40 °C. This MTX-loaded poly (NIPAAm-co-DMAEMA) pH-/thermo-responsive hydrogel can reduce the cell viability of MCF-7 breast cells in comparison with free-MTX treatment [195]. Huang Y et al. synthesized a noncytotoxic pH-/redox-responsive hydrogel by crosslinking the poly(l-lysine isophthalamide) (PLP) and l-cystine dimethyl ester dihydrochloride (CDE). Within this, the PLP can be swollen by stimulation with acid environments or with a reducing agent. Their results suggest that PLP-CDE pH/redox hydrogels may be carriers to provide efficient oral delivery and controlled ion release in intestinal tissue. The stimulated intestinal fluid buffer at pH 6.8 was compared to stimulated gastric fluid (pH 1.2).

The magnesium ions can be released for up to 6 h. In addition, the magnesium ion release can be enhanced in the stimulated intestinal fluid buffer with 1,4-dithiothreitol (DTT). Their results suggest that PLP-CDE pH/redox hydrogels are potential drug carriers to provide efficient oral delivery and control ion release in intestinal tissue [196].

Reactive hydrogel aside, drug delivery can be controlled and released for several hours or days. Among them, thermo-responsive hydrogels have the longest drug release time, but may cause systemic side effects due to their low specificity, especially 37 °C thermo-responsive hydrogels. Although the drug delivery of pH-responsive hydrogels can be adjusted, they are only suitable for specific organs or diseases with unique pH values. Photoresponsive hydrogels can be easily tuned by switching the light energy on or off, resulting in low side effects and adjustable drug concentrations. However, this process requires an external device to provide the light energy. Dual-responsive hydrogels can combine the advantages of both responsive hydrogels. The in vivo effect of dual-responsive hydrogels needs further evaluation. Overall, photoresponsive hydrogels have excellent properties for drug delivery under the available stimulated light energy.

### 4.3. Wound Dressings

The skin is the largest human organ and consists of epidermis, endothelium, and subcutaneous tissue from outside to inside. Skin is attacked by physical, chemical, or thermal damage, which results in wounding. Wounds lead to the destruction of skin structure and function [197]. The creation of a wound will trigger a series of physiological responses that promote wound repair, known as wound healing [198]. Wounds can be categorized by the nature of the repair as acute and chronic wounds. Acute wounds are mainly caused by mechanical injuries, such as abrasions, cuts, burns, scalds, or surgical incisions, and can heal completely in about 8–12 weeks [199]. Chronic wounds are wounds with delayed healing, 12 weeks after the initial injury [194]. These wounds are mainly caused by repeated tissue damage, underlying physiological factors (such as diabetes, impaired angiogenesis, innervation, or cell migration), or acquired physiological factors (such as malignancy or infection) [200,201]. Once a chronic wound forms, it can eventually lead to amputations or even mortality [202]. The wound healing process is dynamic and complicated. It involves four phases: hemostasis, inflammation, proliferation, and remodeling [203]. The hemostasis phase occurs within minutes of the injury. During this period, platelets stick to the wound site, engage with collagen, and release thrombin, which activates fibrin to form a network that stops blood loss. The inflammation phase occurs when immune cells (especially neutrophils and macrophages) are recruited to the wound site by platelets. Immune cells engulf damaged cells, dead cells, bacteria, and other pathogens at the wound site. At the same time, various peptide growth factors are released by platelets and inflammatory cells, which promote the migration of fibroblasts to the wound site.

During the proliferation phase, fibroblasts proliferate at the wound site and rebuild the dermal tissue, employing granulation tissue formation and extracellular matrix protein deposition. Within the granulation tissue, blood vessel networks will be formed, providing sufficient oxygen and nutrients to improve cell survival. Epithelial cells then migrate from the wound edge to the center to cover the defect: this process is termed re-epithelialization. During the remodeling phase, excess collagen fibers are degraded in the dermis, and the wound shrinks and heals rapidly. Therefore, the use of wound dressings to quickly stop bleeding, prevent infection, and promote repair can speed up wound healing and reduce unnecessary mortality.

Characteristics of an ideal wound dressing should (1) provide and maintain a moist environment, (2) permit the easy transmission of gases, (3) remove exudates and absorb blood from the wound, (4) have low adherence to skin, (5) reduce wound necrosis, (6) prevent infection, (7) allow heat insulation, (8) enhance epidermal migration, (9) promote angiogenesis, (10) have low toxicity and be biocompatible and biodegradable [203]. Several studies have shown that hydrogels can form a physical barrier and remove excess exudate. They also provide a moist environment to promote the process of wound healing. In addition, hydrogels can be applied as a sprayable or injectable wound dressing, which may fill irregularly shaped wounds [204,205,206]. They also present with similar properties as the natural extracellular matrix (ECM), biocompatibility, biodegradability, and tunable properties (such as shape, gel state, and mechanical strength). These advantages of hydrogels can simulate the development of hydrogels for different dressings for different types of wounds. Recently, functional hydrogels have received a lot of attention in wound dressing research. These hydrogels can exhibit high-performance biological activities, such as antibacterial properties, promoting blood coagulation, or promoting blood regeneration, etc. [207]. In this section, we focus on the recent application of functional hydrogels for wound dressings.

Wound dressings with hemostasis, angiogenesis, antibacterial infection, and anti-inflammation characteristic have a good impact on wound healing. Natural polymers, such as cellulose, chitosan, collagen, and HA, contain endogenous bioactivation factors. These natural hydrogels are a good wound dressing for wound healing. For example, an in-situ-formed collagen–HA hydrogel was adapted to promote spontaneous wound healing. In addition, this hydrogel inhibited the growth of planktonic *Escherichia coli* (*E. coli*) and *Staphylococcus aureus* (*S. aureus*) [208]. PEG-modified collagen–chitosan hydrogels further reduce the zone diameters of *E. coli* and *S. aureus* biofilms. This hydrogel also exhibits hemostatic ability, which can enhance wound healing [209]. Zhu L. and Chen L. developed CF-encapsulated graphene–silk fibroin macromolecular hydrogel dressings, which have functions of antibacterial (both planktonic and biofilm *S. aureus* and *Pseudomonas aeruginosa* (*P. aeruginosa*)) and enhanced fibroblasts growth. These have a great healing ability for burn wounds [210]. Khaliq T et al. used the chitosan HCl, κ-carrageenan, and PVA-based, physically crosslinked hydrogel to load cefotaxime sodium (CTX), which displays a high oxygen permeability and antibacterial capacity for inhibiting the biofilm size of *S. aureus, P. aeruginosa*, and *E. coli*. This hydrogel provided higher re-epithelialization and good granulation tissue formation for healing burn wounds in a diabetic rat model [211]. In addition, the silver-nanoparticle-loaded pH hydrogel also showed the effective elimination of *P. aeruginosa* and *Staphylococcus epidermidis* (*S. epidermidis*) in in vitro antibacterial biofilm studies. This hydrogel provides a promising strategy to enhance the healing of drug-resistant-bacteria-infected wounds [212]. The in vivo effect of this hydrogel needs further investigation. Collagen–PEG injectable hydrogels containing umbilical cord stem cell factor (SCF) can induce neovascularization and skew toward M2 macrophages in diabetic wounds. They can promote diabetic wound repair based on their angiogenesis and anti-inflammation abilities [213]. It is unknown whether this hydrogel has functions in the inhibition of bacterial growth due to the lack of antibacterial activity assay studies. A 3-carboxy-phenylboronic-acid-grafted gelatin–PVA hydrogel exhibits excellent hemostasis properties enhancing cell adhesion. This hydrogel further encapsulates the vancomycin-conjugated silver nanoclusters (VAN-AgNCs) and nimesulide (NIM), endowing an anti-inflammatory effect. It also has the capacity to inhibit the planktonic *S. aureus* and *P. aeruginosa* growth in a VAN-AgNCs dose-dependent manner. In an in vivo experiment, this VAN-AgNCs- and NIM-loaded 3-carboxy-phenylboronic-acid-grafted gelatin–PVA hydrogel can induce the sequential healing processes to promote the healing of chronically infected diabetic wounds [214]. Plasma-exosomes-loaded, pH-responsive carboxymethylcellulose (P-Exos-loaded CMC) hydrogel stimulates the activation of the vascular endothelial growth factor (VEGF) signaling pathway. This pathway further enhances angiogenesis and re-epithelialization to promote the wound healing process in diabetic type 1 mice [215]. Another study uses the umbilical-cord-derived mesenchymal stem cell exosomes combined with Pluronic F127 hydrogel to demonstrate that this hydrogel can induce the expression of transforming growth factor beta-1 (TGFβ-1) and cell proliferation in addition to VEGF production. Based on the above ability, it can enhance the regeneration of granular tissue and angiogenesis in chronic diabetic wound healing [216]. However, the antibacterial activity of exosomes-loaded hydrogels is unclear in these two studies [215,216].

Oxidative stress is one of the factors which leads to impairment of wound healing. Hydrogels with antioxidant capacity have also been developed as wound dressings. Xu J et al. prepared a thermo-responsive hydrogel using a combination of PEG, polypropylene glycol (PPG), and polydimethylsiloxane (PDMS), which then incorporates the lignin (an antioxidant material). This hydrogel increases cell proliferation to promote the healing of burn wounds [217]. Lignin have been reported to inhibit growth of Gram-positive bacteria (*Listeria monocytogenes* (*L. monocytogenes*) and *S. aureus*) and yeast (*Candida lipolytica* (*C. lipolytica*)) [218]. This may reveal that lignin thermo-responsive hydrogels have an inhibitory effect on bacterial growth. Zhang J et al. have developed an antibiofouling hydrogel, which is composed of, and involves the balancing of, oppositely charged alginate, hyaluronic acid (HA), and polylysine (PLL). This antibiofouling hydrogel can resist protein adhesion to avoid the immune response. It can also resist the adhesion of *E. Coli* and *S. aureus*, which reduces the occurrence of bacterial infection. In addition, this hydrogel presents anti-inflammation, ROS elimination, and angiogenesis promotion abilities via incorporating curcumin (Cur) and epigallocatechin gallate (EGCG). This Cur-EGCG antibiofouling has a good effect in the treatment of ionizing-radiation-induced skin injury [219]. Another study utilities polydopamine to endow hydrogels with antioxidant ability. These hydrogels can reduce AgNO_3_ into Ag nanoparticles (AgNPs) and then show an antibacterial activity on planktonic *E. coli* and *S. aureus* growth [220]. However, there is a need to further determine the in vivo effect of polydopamine hydrogels in wound healing.

Scientists have developed functional hydrogels using different polymers and bioactive factors, demonstrating their potential application as wound dressings. Thus, the question of how to adjust and expand the functions of hydrogels will be a direction to consider in the future.

### 4.4. Tissue Engineering

Tissue engineering is a promising and challenging strategy to treat patients who suffer functional failure and irreparable tissue destruction [221]. The aim of tissue engineering is to develop a scaffold mimicking an in vivo extracellular matrix to support tissue regeneration. Hydrogels have gained great interest in tissue engineering due to their mechanical strength, biocompatibility, biodegradability, and the resemblance to in vivo extracellular matrix [222].

A hydrogel scaffold can be useful in tissue regeneration of nerves, cardiac tissue, cartilage, and bone. For example, the 3D printing of collagen–chitosan is beneficial in decreasing scar and cavity formation and can improve the regeneration of nerve fibers, as well as functional recovery, when tested in an animal model [223]. Another example is HA combined with alginate and fibrin. This was applied as an ink ingredient of 3D printing in peripheral nerve tissue regeneration [224]. In addition, the HA–cellulose hydrogels can repair the central nerves [225]. Li J et al. used horseradish peroxidase (HRP) and choline oxidase (ChOx) crosslinked gelation hydrogel to encapsulate the mMSCs. This hydrogel displays a high capacity to promote cellular viability, neural differentiation, and neurotrophic secretion of loaded mMSCs. Based on that capacity, it can enhance the survival and proliferation of endogenous neural cells and neurological function recovery of traumatic-brain-injured mice [226].

Previous studies indicated that hydrogels may promote the differentiation of hBMSC to nucleus pulposus cells [227]. Finklea FB et al. have developed the hiPSCs-encapsulated PEG–fibrinogen microsphere hydrogel, which can support efficient cardiac differentiation and produce cardiomyocytes. This hydrogel model displays a potential role in injection-based regenerative therapies [228]. Long G et al. has used the alginate (A)-/silk sericin (SS)-lamellar-coated antioxidant system (ASS@L) to encapsulate the adipose-derived stem cells (ADSCs). ADSCs-ASS@L is an injectable hydrogel. In their study, ADSCs- ASS@L has favorable effects in cardiac damage therapy in acute myocardial infarction. Kim KS et al. has precisely adjusted the 3D environment of encapsulated cells to increase graft survival. They constitute a porous mesh structure loaded with human cardiomyocytes, human cardiac fibroblasts, and a gelatin–methacryloyl–collagen hydrogel. This 3D cardiac mesh (cMesh) tissue has a high ability to maintain the cell viability of encapsulated cells and improve the cardiac function of rats with acute myocardial infarction [229]. From the in vitro cardiac study, silk-fibroin can also induce pacemaker cells to have functional and morphological characteristics of genuine sinoatrial-node cardiomyocytes. In addition, these pacemaker cells generated by injection of silk fibroin in the left ventricles of rats acted as a in situ sinoatrial node [230].

Wang G et al. have reported that chondro-spheroid gelatin methacrylate (GelMA), crosslinked with hyaluronic acid methacrylate (HAMA) hydrogel, can enhance the cell proliferation, aggregation, and morphology in vivo. Their results suggest that this 3D cell-laden tissue may have a potential role in cartilage tissue engineering [231]. The gelation can also be combined with carboxymethyl cellulose (CMC) and alginate as bioinks to generate the human knee meniscal scaffold by 3D printing. This scaffold can increase collagen secretion and cellular proliferation of MG-63-osteosarcoma cells. It indicates that this scaffold is suitable for cartilage tissue engineering [232]. Yan J et al. indicate that the rMSCs-laden hydroxyapatite–collagen type I (HAC) and PLGA-PEG-PLGA thermogel can repair the femoral condyle defect in rabbit [233]. Shi W et al. show that dynamic hyaluronic acid hydrogel with covalent-linked gelatin can reduce the reactive oxygen species for influence of cartilage tissue regeneration [234].

Bordini EAF et al. report that the Dex-loaded nanotube-modified gelatin hydrogel can induce bone regeneration in an inflammatory microenvironment [235]. These 3D-printed gelatin/sodium alginate hydrogel scaffolds, doped with nano-attapulgite, can provide the high structural complexity of bone. This scaffold can facilitate the osteogenesis of mMSCs and repair the rabbit tibia plateau defect [236].

Many natural, synthetic, or semi-synthetic hydrogel scaffolds can effectively repair nerves, heart, and soft and hard tissues through animal experiments. However, these studies are limited to preclinical trials, and their clinical value may require further research.

## 5. Conclusions

This review introduces the classification, structure, and recent applications of hydrogels in biomedical fields such as 3D cell cultures, drug delivery, wound dressings, and tissue engineering. Semi-synthetic hydrogels can simultaneously optimize the disadvantages of natural and synthetic hydrogels in 3D cell cultures and mimic the extracellular matrix of living tissue. These hydrogels have an excellent ability to comprise 3D cell culture scaffolds. In addition, combined with 3D printing technology, these 3D scaffolds have recently been applied in tissue engineering. However, the clinical value of these 3D scaffolds in tissue engineering should be further evaluated. In drug delivery, the photoresponsive hydrogel has excellent properties for drug delivery under the available stimulated light energy. However, utilization of other materials or other devices may lift the restriction of other responsive hydrogels. Functional hydrogels rapidly contribute the wound healing due to their bioactive properties in healing processing. The multifunctional hydrogel may be a promising strategy due to the complex and dynamic progression of wound healing. However, very few hydrogels are commercially available to be applied in clinical treatment in 3D scaffolds, drug delivery, and tissue engineering [237]. Thus, understanding the extracellular matrix mechanical strength, elasticity, and biological composition of each cell and tissue will broaden those medical applications of hydrogels and expand the use of hydrogels for clinical treatment.

## Figures and Tables

**Figure 1 molecules-27-02902-f001:**
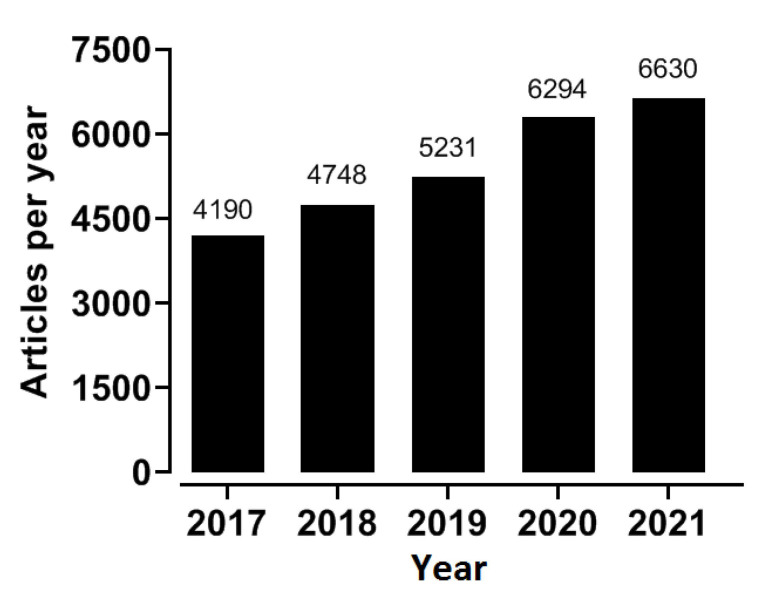
The number of publications related to hydrogels in the biomedical field in the past five years. The data was obtained from the biomedical database PubMed, using the search term “hydrogel”.

**Figure 2 molecules-27-02902-f002:**
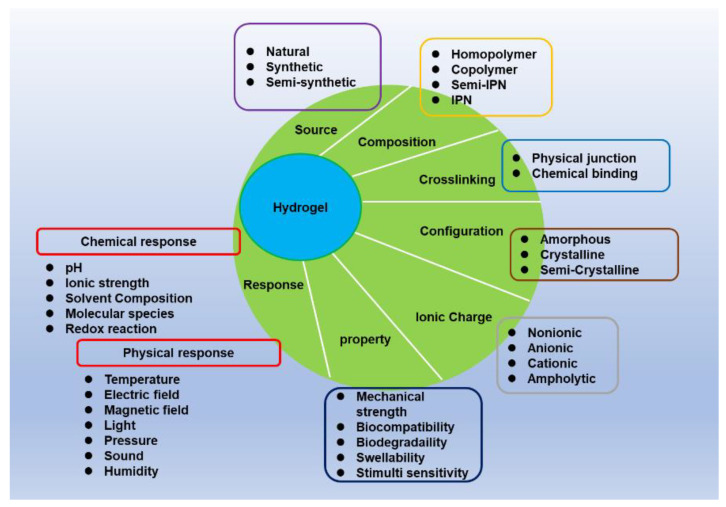
Classification of hydrogels.

**Figure 3 molecules-27-02902-f003:**
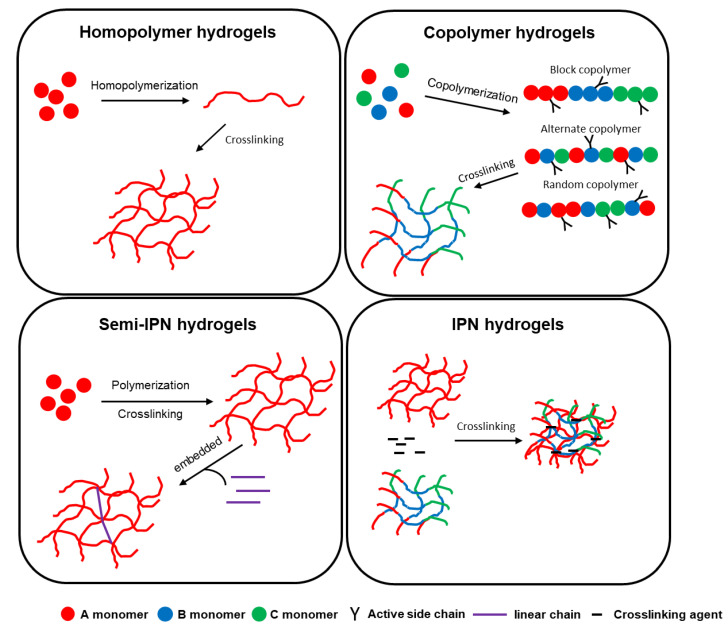
Simple diagram of homopolymer hydrogels, copolymer hydrogels, semi-IPNs, and IPN hydrogels.

**Figure 4 molecules-27-02902-f004:**
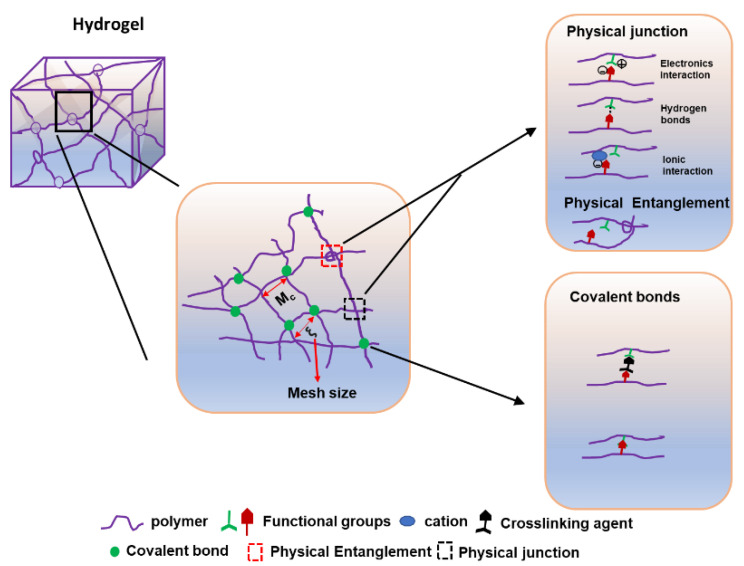
The structure of hydrogels (chemical linking and physical junctions).

**Figure 5 molecules-27-02902-f005:**
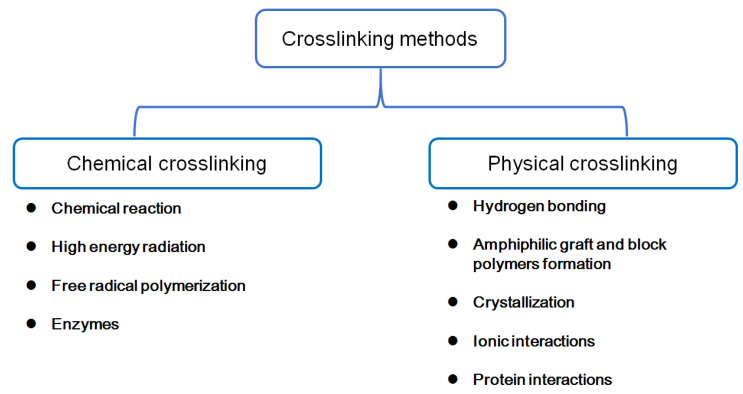
Methods of chemical and physical crosslinking for hydrogels preparation.

**Table 1 molecules-27-02902-t001:** Natural, synthetic, and semi-synthetic hydrogels for 3D cell cultures.

Source of Hydrogels	Properties	Materials	Cell	Applications
Natural	Provide comparable viscoelasticity and fibrils to the ECM; having good biocompatibility; endogenous factors can support cellular activity	Collagen	Rat chondrocyte [127], hMSCs [128,136], rMSC [129], HUVECs/hASCs [133]	Maintain the chondrocyte phenotype [127]; facilitate chondrogenic differentiation of hBMSCs [128] and rBMSCs [129]; form stable EC networks [133]; promote cell viability; promote growth of hMSCs [136].
		HA	hiPSC-NPCs [141], hiPSCs [142], rMSCs [143], human breast cancer MCF-7 cells [146], HepG2 cells [147], human dental pulp stem cells [148], hNS/PC [149], and hMSCs [150]	Promote the neural differentiation of hiPSC-NPCs [141]; cardiac differentiation of hiPSCs [142]; osteogenic differentiation of human dental pulp stem cells [148]; the adhesion and proliferation of HepG2 cells [147]; cell spreading, fiber remodeling, and focal adhesion of hMSCs [150]; maintain the stemness of rMSCs and induce the direct cartilage differentiation [153]; enhance the tumorigenic capability of MCF-7 cells [146]; increase the oligodendrocytes and neural differentiation of hNS/PC and support long-term cell viability [149].
		Fibrin	HUVECs/hMSCs [154], porcine cumulus–oocyte complexes (COCs) [155], primary human chondrocytes [156], mHPSCs [157], and hiPSCs/HUVECs/human dermal fibroblast [158]	Prevascular formation of HUVECs, improve cell viability and proliferation of hMSCs and enhance their osteogenic differentiation and bone mineral deposition [154]; maintain the functional relationship between oocytes and follicular cells [155]; induce the production of glycosaminoglycans and collagen type II of primary human chondrocytes [156]; enhance the murine hematopoietic stem/progenitor cells (mHPSCs) expansion and differentiation [157]; no effect viability and prevascular formation of encapsulated cells [158].
		Alginate	hESCs/hiPSCs [162], hiPSCs-derived neurons [163]	Enhance the generation of retinal pigmented epithelium and neural retina of hESCs/hiPSCs [162]; form complex neural networks [163].
Synthetic	Have the good mechanical strength to provide structural support for various cell types in 3D cell culture	PVA	mHSCs [166], mSCCs [167], human glioma cell lines LN299, U87MG and Gli36 [168], human breast cancer Hs578T cells, and human pancreatic cancer cell lines Sui67 and Sui72 [169]	Enhance the expansion of murine hematopoietic stem cells (mHSCs) [166]; promote the meiotic and post-meiotic differentiation rate of mSCCs [167]; form tumor spheroids [168,169].
		PEG	hiPSCs [171], mMSCs [172], chondrocyte [174], and hMSCs [176]	Enhance the hematopoietic differentiation of hiPSCs [171]; evaluate the behavior of mMSCs [172] and hMSCs at the specific condition [176]; prolong the oxygen release of chondrocytes [174].
Semi-synthetic	Have a feature of ECM microenvironment and faster stress relaxation	HA–PEG	hiPS-HEPs [177] and HUVECs [178]	Enhance viability and functionality of hiPS-HEPs [177]; promote the capillary-like sprouts formation of HUVECs spheroids [178].
		RGD–alginate–PEG	Fibroblasts and mMSCs [179]	Increase the spread and proliferation of fibroblasts and the osteogenic differentiation of mMSCs [179].

**Table 2 molecules-27-02902-t002:** Smart hydrogels for drug delivery.

Hydrogels	Drug	Materials	Sustained-Release Time	Proposed Applications	Ref.
Thermoresponsive hydrogel	Dexamethasone	HPMA	More than 30 days	Osteoarthritis and rheumatoid arthritis	[181]
	Topotecan	Poloxamer 407 and poloxamer 188	28 days	Colorectal cancer	[183]
	Lamivudine and zidovudine	Pluronic^®^ F-127	168 h	AIDS	[184]
	Antibody	PEGMA	13 days	Enhance the efficacy of antibody treatment	[185]
pH-responsive hydrogel	Bortezomib	mPEG-LUT	50 h	Colorectal cancer	[188]
	Amifostine (S-2(3-aminopropylamino) ethyl phosphorothioate	MAC-g-PCL	6 h	Acute radiation syndrome	[189]
Photoresponsive hydrogel	Doxycycline	SPCOOH modified-silicone-hydrogel (poly(HEMA-co-PEGMEA))	42 h	Inflammation disease	[192]
	Insulin	BP, pNIPAM, PEG, and ETPTA	Not detected	Diabetic disease	[193]
Daul-responsive hydrogel					
pH/thermo	Doxorubicin chemosensitizer curcumin	poly (NIPAAm-co-DMAEMA)	168 h	Colon cancer	[194]
	Methotrexate		50 h	Breast cancer	[195]
pH/redox	Magnesium ions	poly (NIPAAm-co-DMAEMA) PLP-CDE	6 h	Ionic therapeutics	[196]

## Data Availability

The data that support the findings of this study are contained within the article. More information is available on request from the corresponding author.

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
