# Peer review of "Hydrogels: Properties and Applications in Biomedicine"

_molecules, 2022, doi:10.3390/molecules27092902_

Round 1

Reviewer 1 Report

The authors have presented a very detailed review on hydrogels in this manuscript, which will be very useful to readers who are new to this research topic. The manuscript is well written and easy to understand. A few minor concerns are listed below.

  1. On page 1, in Line 36, hydrogels are associated with biodegradability and biocompatibility. It should be noted that not all hydrogels are biocompatible and biodegradable. For instance, synthetic hydrogels such as polyacrylamide hydrogels are not biodegradable. Also, it is difficult to understand what “design ability” refers to.
  2. On Page 7, in Table 1, authors may consider including additional hydrogels such as fibrin.

Author Response

The authors have presented a very detailed review on hydrogels in this manuscript, which will be very useful to readers who are new to this research topic. The manuscript is well written and easy to understand. A few minor concerns are listed below.

  1. On page 1, in Line 36, hydrogels are associated with biodegradability and biocompatibility. It should be noted that not all hydrogels are biocompatible and biodegradable. For instance, synthetic hydrogels such as polyacrylamide hydrogels are not biodegradable. Also, it is difficult to understand what “design ability” refers to.

Reply: Thank you for your suggestion. It has been noted and reedited the sentences on page 1, in Line 35. Please see the modification as below (on page 1 in Line 35 to Line 39).

“Although not all hydrogels have good biodegradability and biocompatibility such as synthetic hydrogels (polyvinyl alcohol, polyacrylamide, polyethylene glycol hydrogels, etc.), their biodegradability and biocompatibility can be adjusted by modification of their functional group or incorporation of natural polymers. Hence, scientists are still interested in the biomedical application of hydrogel due to its adjustable creativity for different uses.”

  1. On Page 7, in Table 1, authors may consider including additional hydrogels such as fibrin.

Reply: Thank you for your suggestion. We included the fibrin hydrogels in recent application of 3D cell culture as described below. (Page 9, Line 336-362) “Fibrin is also one major protein to construct ECM in addition to collagen and HA, which are formed by fibrinogens [162]. Fibrin hydrogel presents the bioactivities such as angiogenesis, and cell-matrix interaction through fibrin clotting, which have been used for 3D cell culture of adipose-derived stem cells and cardiomyocytes [163, 164]. Recent studies of fibrin hydrogels are associated with mimicking a native ECM matrix and evaluating its effect on 3D cell culture. Heo D.N. et al. mimicked a native ECM matrix by incorporating the collagen in fibrin hydrogels. The collagen/fibrin hydrogels can induce the pre-vascular formation of HUVECs which further improves the cell viability and proliferation of hMSCs and promotes their osteogenic differentiation and bone mineral deposition [165]. Gorczyca G. et al. employed an alginate/fibrin IPN hydrogel which can form a gas and nutrition exchange space as a native ECM. They found that this hydrogel can encapsulate the porcine cumulus-oocyte complexes (COCs) and maintain the functional relationship between oocyte and follicular cells. They provided a novel culture platform for oocyte in vitro maturation which is important in reproductive biotechnology [166]. Bachmann B. et al. adjusted the stiffness of fibrin hydrogels toward the elasticity of native chondrocyte ECM by using different concentrations of fibrin. Their study showed that the fibrin hydrogel under 30 kPa Young’s best modulus can induce the production of glycosaminoglycans and collagen type II from 3D cultured-primary human chondrocytes. It is best to trigger the chondrocyte redifferentiation [167]. Another study also used a similar method to mimic the elasticity of the bone marrow niche. Their results show the fibrin hydrogel prepared under lower pressure as a 3D scaffold is the best to enhance the murine hematopoietic stem/progenitor cells (mHPSCs) expansion and differentiation [168]. However, the fast biodegradability of fibrin hydrogels results in an unstable 3D cell culture environment. Jarrell D. K. et al. found that biodegradable times of high salt fibrin hydrogel are lower than physiologic-salt fibrin hydrogel. This hydrogel did not affect the viability and pre-vascular formation of encapsulated cells such as human dermal fibroblasts, hiPSCs, and HUVECs [169].”

Reviewer 2 Report

  1. 44-45Hydrogels are not only based on polymers but are defined as following: Gel in which the swelling agent is water. The network component of a hydrogel is usually a polymer network.A hydrogel in which the network component is a colloidal network may be referred to as an aquagel.
  2. 49 Properties is quite vague please add more explanation
  3. I miss a story in the introduction. This should be better structured
  4. Part 2 Structure and classification of hydrogels : This should be explained in scientific correct way and not like its explained right now. I would use figure 3 as a solid base to start this chapter.
  5. 75-76The crosslinking of hydrogels can be 75 formed by the covalent (chemical) or physical (junction/entanglement) linking. This should be explained more and more attention should be given to this
  6. This also needs better structuring, first Mesh size is discussed subsequently Mw and there is no explanation on how this is in relation between each other
  7. 85 what about supramolecular polymers which use functional groups to cross link?
  8. 104-105 How does the author explains that there is inherent biocompatibility while there is an immunological reaction?
  9. 111 not all of these polymers are biocompatible? It heavily depends on the definition of biocompatibility
  10. 125 Copolymers are indeed polymers that are made out of two or more monomers however the simplified diagrams loses information, what about block, random copolymers? You can also have copolymers with an active side chain which is able to form the cross links.
  11. 20 in vivo and in vitro should be in italic
  12. 248 collogen should be collagen
  13. There are major developments in tissue engineering on Gelatine. This is not even mentioned. This should be added
  14. Why is opsonisation not discussed in the drug delivery part? This is major concern
  15. 534 the four stages of wound healing should be discussed
  16. 562-565 I would not add such a statement as each type of wound needs a different dressing. So this should be adapted
  17. Antibacterial capacity should be specified for each example in the wound dressing. Is this bactericidal of does it inhibit the growth. This is a major difference next to the fact if the bacteria are planktonic or they are found in a biofilm
  18. This review wants to explain everything while not saying a thing. Only 140 references are added, this is not enough!

Author Response

  1. 44-45Hydrogels are not only based on polymers but are defined as following: Gel in which the swelling agent is water. The network component of a hydrogel is usually a polymer network.A hydrogel in which the network component is a colloidal network may be referred to as an aquagel.

Reply: Thank you for your suggestion. We added the definition of hydrogel as following (Page 1, line 47-51): Hydrogel is a three-dimensional (3D) network, which can absorb a large amount of water and swell in the water due to their hydrophilic groups such as -NH2, -COOH, -OH, -CONH2, - CONH, and -SO3H. Its network is usually constructed by cross-linked polymer chains that sometimes can be formed through cross-linked colloidal clusters.

  1. 49 Properties is quite vague please add more explanation

Reply: Thank you for your suggestion. We added that as following (Page 2, line 61-64): Along with the properties (e.g, swelling-deswelling rate, stiffness, degradability, mech size) of hydrogels can be adjusted by the changing of hydrophilic and hydrophobic ratios, initiator or polymer concentration, reaction conditions (time, temperature, container, etc.)

  1. I miss a story in the introduction. This should be better structured

Reply: Thank you for your suggestion. The architecture of introduction has been modified. Please see the introduction part (Page 2).

  1. Part 2 Structure and classification of hydrogels: This should be explained in scientific correct way and not like its explained right now. I would use figure 3 as a solid base to start this chapter.

Reply: Thank you for your suggestion. We reorganized the part 2 structure and classification of hydrogels and used the figure 3 (now is figure 2) to start this chapter.

  1. 75-76The crosslinking of hydrogels can be 75 formed by the covalent (chemical) or physical (junction/entanglement) linking. This should be explained more and more attention should be given to this

Reply: Thank you for your suggestion. We explained as following (Page 5, Line 187-191): As figure 4 (bellow), the covalent bond can be linked between two functional groups of polymers with or without covalent agent (under the enzyme catalysis). The physical junction can be formed by the electronic interactions between the opposite charges in the polymer. It also can be linked with hydrogen bonds or ionic interactions with the ions in solution.

  1. This also needs better structuring; first Mesh size is discussed subsequently Mw and there is no explanation on how this is in relation between each other

Reply: Thank you for your suggestion. We structured and explained the relation between Mw and mesh size as following (Page 5, Line 194-201):” The size of mesh hole is defined as mesh size, which is associated with the release time of drug delivery [101] and correlated with the linear distance between two crosslinking points (ξ) [101,102]. Demeter M. et al. used the rheological analysis to demonstrate that the average molecular weight between crosslinking points (Mc) and crosslink density (Ve) affects the ξ of mesh size. There is a positive correlation between Mc and mesh size [103]. Steinman NY. Et al. demonstrated that 4 kDa and 8 kDa PEG can form the 11 kDa and 16 kDa hydrogels at the same concentration of crosslinking agent [104].”

  1. 85 what about supramolecular polymers which use functional groups to cross link?

Reply: Thank you for your suggestion. We explained as following (Page 5, Line 179-186): Supramolecular polymers are a new category of polymers that can potentially be used for bio-material applications beyond the limitations of conventional polymers. The molecular weight of supramolecular hydrogels may not be infinite due to their crosslinking is a series of specific but non-permanent interactions [97-100]. They are polymeric arrays of monomeric units held together by reversible and highly directed secondary interactions, which are non-covalent bonds, such as hydrogen bonding interaction, π-π stacking, host-guest interaction. Thus, the resulting materials retain their polymeric properties in solution.

  1. Zhang G, Chen Y, Deng Y, Ngai T, Wang C. Dynamic Supramolecular Hydrogels: Regulating Hydrogel Properties through Self-Complementary Quadruple Hydrogen Bonds and Thermo-Switch. ACS Maccro Letters. 2017;6(7):641-646
  2. Li J, Li X, Ni X, Wang X, Li H, Leong KW.Self-assembled supramolecular hydrogels formed by biodegradable PEO-PHB-PEO triblock copolymers and α-cyclodextrin for controlled drug delivery. Biomaterials. 2006;27:4132-4140
  3. Grindy SC, Learsch R, Mozhdehi D, Cheng J, Barrett DG, Guan Z, Messersmith PB, Holten-Andersen H. Control of hierarchical polymer mechanics with bioinspired metal-coordination dynamics. Nat. Mater. 2015;14:1210-1216
  4. Rodell CB, Mealy JE, Burdick JA. Supramolecular guest-host interactions for the preparation of biomedical materials. Bioconjug. Chem. 2015;26:2279-2289

  1. 104-105 How does the author explains that there is inherent biocompatibility while there is an immunological reaction?

Reply: We want to mention that" Although natural hydrogels are safe for a lot of population, some materials of natural hydrogels are allergens in rare cases [71-76]. Thus, above natural hydrogel has potential immunological risk if they are used in sensitive individuals.." (Page 3, line 91-93) We also add these references in line 93.

  1. Wan Ali WNS, Ahmad Tarmidzi NA. A Rare Case of Contact Allergy towards Impression Compound Material. Eur J Dent. 2021;15(4):798-801.
  2. Janas-Naze A, Zhang W. Perioperative anaphylaxis to fibrin sealants in children with Noonan Syndrome - a retrospective study. Ann Allergy Asthma Immunol. 2022;S1081-1206(22)00206-X.
  3. Coury JR, Skaggs KF, Marciano GF, et al. Intraoperative Anaphylaxis to the Bovine Flowable Gelatin Matrix: A Report of 2 Cases. JBJS Case Connect. 2022;12(1):10.2106/JBJS.CC.21.00753.
  4. Shimojo N, Yagami A, Ohno F, et al. Fish collagen as a potential indicator of severe allergic reactions among patients with fish allergies. Clin Exp Allergy. 2022;52(1):183-187.
  5. Alawami AZ, Tannous Z. Late onset hypersensitivity reaction to hyaluronic acid dermal fillers manifesting as cutaneous and visceral angioedema. J Cosmet Dermatol. 2021;20(5):1483-1485.
  6. Eshraghi B, Shaygan P, Lajevardi N, Fazel M. Nicolau syndrome, a rare though important complication of hyaluronic acid injection. Int J Dermatol. 2021;60(7):e285-e286.

  1. 111 not all of these polymers are biocompatible? It heavily depends on the definition of biocompatibility

Reply: Thank you for your suggestion. We revised the sentence as "Although a few of them are biocompatible such as PAAM, they are stable and have mechanical strength." In Page 3, line 99-100.

  1. 125 Copolymers are indeed polymers that are made out of two or more monomers however the simplified diagrams loses information, what about block, random copolymers? You can also have copolymers with an active side chain which is able to form the cross links.

Reply: Thank you for your suggestion. We added the block and random copolymers with active side chains. Please see figure 3.

  1. 20 in vivo and in vitro should be in italic

Reply: Thank you for your suggestion. In vivo and in vitro have been modified as italic.

  1. 248 collogen should be collagen

Reply: Thank you for your suggestion. It has been adapted. Please see line 269.

  1. There are major developments in tissue engineering on Gelatine. This is not even mentioned. This should be added

Reply: Thank you for your suggestion. But we have mentioned the developments in tissue engineering on gelatine by using the other universal name of gelatine as ‘gelatin’.

  1. Why is opsonisation not discussed in the drug delivery part? This is major concern

Reply: Thank for your suggestion. We add the reason why did not discuss the opsonization. Please see Page 12, line 432 to 450.

“The benefit from the stimuli-responsive property can enable novel targeted drugs and control drug release through non-intravenous administration. It can also delay the effect of opsonization by low blood contact.

The basic advantage is the ability of a smart hydrogel to change its properties (such as mechanical properties, swelling capacity, hydrophilicity, or permeability of bioactive molecules) under the effect of surroundings, including temperature, pH, electromagnetic radiation, magnetic field, and biological factors. Smart hydrogels can be prepared by the natural or synthetic polymers. There is a problem with natural hydrogels, which is that its mechanical properties make it difficult to maintain consistency.

Although this problem with natural hydrogel can be overcome by extensive chemical modification for natural polymers, it is very difficult to process [192]. In contrast, synthetic polymers are easy to alter their chemical or physical properties. The biodegradable and hydrophilic synthetic polymers are the most competitive substances for synthesis of the smart hydrogel for drug delivery. Those synthetic polymers endow the smart hydrogel with low toxicity, low side effect and low blood material adhesion. Of which, the low blood material adhesion can slow the effect of opsonization and reduce the phagocyte elimination. Table 2 shows the application of those synthetic smart hydrogels for drug delivery.”

  1. 534 the four stages of wound healing should be discussed

Reply: Thank you for your suggestion. In this review, the four stages of wound healing have been discussed in Page 17. We have changed the names of the four phases of wound healing as “hemostasis, inflammation, proliferation, and remodeling”. Please see the modifications in lines 599 to 620.

  1. 562-565 I would not add such a statement as each type of wound needs a different dressing. So this should be adapted

Reply: Thank you for your suggestion. We organized and adapted the statement. Please see the Page 17, line 634 to line 642.

Several studies have been showed that hydrogels can form a physical barrier and remove excess exudate. They also provide a moist environment to promote the process of wound healing. In addition, the hydrogel can be applied as a sprayable or injectable wound dressing, which may fill irregularly shaped wounds [206-208]. They also present similar properties to a natural extracellular matrix (ECM), biocompatibility, biodegradability, and tunable properties (such as shape, gel state, and mechanical strength). Due to those advantages of hydrogels, which increase the development of hydrogels on different dressings for each type of wound.

  1. Antibacterial capacity should be specified for each example in the wound dressing. Is this bactericidal of does it inhibit the growth. This is a major difference next to the fact if the bacteria are planktonic or they are found in a biofilm

Reply: Thank you for your suggestion. We added the antibacterial activity of each example. Please see the part of wound dressing.

  1. This review wants to explain everything while not saying a thing. Only 140 references are added, this is not enough!

Reply: Thank you for your suggestion. We have included more references. Please see the reference list.

Reviewer 3 Report

This is a review of hydrogels in biomedicine. Hydrogels are promising and interesting research target among biomedical scientists. So there are already many review papers about this topic. I think this review lack novelty. Here are some comments:

1, The authors cite a lot of reviews with similar topic. I would say this paper is just a summary of reviews while not research articles. The authors should summarize the research articles and based on that, offer their own opinions about hydrogels.

2, Line 84-95 seems to be a summary of reference 10, which is also a review of hydrogel. The authors should go to find the original article and cite those articles.

3, Line 141, the authors even didn’t cite the original paper but the review paper.

4, Figure 2 is a reprinting of Figure 1 in reference 14, Figure 3 is pretty close to Figure 2 in reference 14. 

Author Response

This is a review of hydrogels in biomedicine. Hydrogels are promising and interesting research target among biomedical scientists. So there are already many review papers about this topic. I think this review lack novelty. Here are some comments:

1, The authors cite a lot of reviews with similar topic. I would say this paper is just a summary of reviews while not research articles. The authors should summarize the research articles and based on that, offer their own opinions about hydrogels.

Reply: Thanks for your suggestion. Due to some of the information being elementary knowledge, we add a few reviews as references for the source of natural and synthetic hydrogels, stage of wound healing, characteristics of an ideal wound dressing, and challenging strategy of tissue engineering. Apart from that information, we cited the original references.

2, Line 84-95 seems to be a summary of reference 10, which is also a review of hydrogel. The authors should go to find the original article and cite those articles.

Reply: Thanks for your suggestion. We found the original articles and cited those articles.

3, Line 141, the authors even didn’t cite the original paper but the review paper.

Reply: Thanks for your suggestion. We have cited the original paper.

4, Figure 2 is a reprinting of Figure 1 in reference 14, Figure 3 is pretty close to Figure 2 in reference 14.

Reply: Thank you for your suggestion. We edited figure 3, and then rearranged the as figure 2 following another reviewer’s suggestion. The figure has been modified.

Round 2

Reviewer 2 Report

This can be submitted

Reviewer 3 Report

The authors did some corrections and now this paper can be accepted.